# Efficient phrase search with reliable verification over encrypted cloud-IoT data



Wanshan Xu[1], Ze Zhu[1] and Muhammad Irfan Khalid[2]

[1] School of Computer and Cyberspace Security, Communication University of China, Beijing, China

[2] Faculty of Computing and Information Technology, Department of Information Technology, University of Sialkot, Sialkot, Pakistan

## ABSTRACT

Phrase search encryption enables users to retrieve encrypted data containing a sequence of consecutive keywords without decrypting, which plays an important role in cloud Internet of Things (IoT) systems. However, due to the sequential relationship between keywords in the phrase, phrase search and verification are more difficult than multi-keyword search. Furthermore, verification evidence is generated by the server in existing schemes, and cloud servers are generally considered untrustworthy, so the verification is unreliable. To address this, we propose an efficient phrase search scheme that supports reliable verification of search results, where blockchain is introduced to generate verification evidence and perform verification of the results. The immutable nature of blockchain guarantees the credibility of evidence and verification. During the verification, we use a multiset hash function to generate aggregated evidence, reducing storage and blockchain transaction costs. In addition, we design a novel composite index and discrimination algorithm based on homomorphic encryption, with which we can quickly identify phrases and improve search efficiency. Finally, we conducted security analysis and detailed experiments on our scheme, which proved that the scheme is secure and efficient.

## INTRODUCTION

Currently, the Internet of Things (IoT) has developed rapidly and is widely used in agriculture, industry, medicine and other fields, helping to improve crop production, manufacturing efficiency, and protect patients' health. Every day, hundreds of millions of IoT devices around the world generate massive amounts of data, which is stored on the local or cloud. Compared to local storage, cloud storage can not only reduce local storage and management costs, achieve efficient data processing and analysis, but also help to achieve data sharing between different users, so it has been widely researched and applied.

Although cloud storage brings many conveniences to users, it also poses security and privacy risks. Cloud servers are generally considered untrustworthy, the unauthorized inside user may attempt to access sensitive information (*e.g.*, patient's disease name, blood pressure, *etc.*), and some hackers may also illegally access data, which will lead to data

Corresponding author
Wanshan Xu,
xuwanshanxws@126.com

destruction and privacy leaks. In this case, IoT devices generally encrypt data first, and then outsource the ciphertext to the cloud to protect the integrity and privacy of the data.

For data outsourced to the cloud, when users need to access it, they perform retrieval on ciphertext. To achieve keyword retrieval on encrypted data and maintain the balance between search efficiency and security, *Song, Wagner & Perrig (2000)* proposed the concept of searchable encryption (SE), according to the number of keywords queried, SE is divided into two categories: single keyword search and multi-keyword search. Phrase search is an important technology of searchable encryption, which can search for a series of conjunction keywords in sentences or documents (*Tang et al., 2012*; *Anand et al., 2014*). Designing an efficient phrase search solution is very challenging, existing single keyword (*Curtmola et al., 2006*; *Stefanov, Papamanthou & Shi, 2014*) or multi-keyword encryption search schemes (*Cash et al., 2013*; *Poon & Miri, 2015*) cannot be directly applied to phrase search because they cannot determine the location of keywords. For example, in the electronic medical system, certain diseases are expressed by phrases, such as "myocardial infarction". When searching for this phrase with a multi-keyword encrypted search scheme, the cloud server may return search results that contain both "myocardium" and "infarction", but they may not appear as a phrase. Obviously, the search results contain a lot of invalid files.

Another challenge for phrase search is the verification of search results. Since data is outsourced on the cloud, external or internal attacks on cloud server may compromise the integrity or confidentiality of the data. In addition, data may be lost or damaged during data transmission. Therefore, it is necessary to verify the results of phrase search.

Although there are some studies (*Kissel & Wang, 2013*; *Ge et al., 2021*) addressing the problem of phrase search result verification, unfortunately, these verification schemes lack reliability. The reason is that in the existing solution, the server calculates search results and uses methods such as RSA accumulators to generate verification evidence. These search results and verification evidence may be forged by the cloud server (for example, the server may store only a part of the file and search index for financial gain, in which case the search results and verification evidence are incomplete). In addition, data users may forge verification results for cost savings, which may also result in the unreliability of verification results. In recent studies, some researchers have adopted blockchain technology. These schemes guarantee the reliability of verification based on the immutable property of the blockchain and have obtained ideal experimental results. But, these schemes mainly focus on the encrypted search of single keyword and cannot be applied to phrase search.

To address these problems, we design a blockchain-based phrase search scheme supporting reliable verification over encrypted cloud-IoT data, our main contributions are as follows:

1) We propose an efficient phrase search scheme over encrypted cloud-IoT data. In our scheme, a composite index containing keyword position and a distance discrimination algorithm based on homomorphic encryption are proposed, which can not only reduce the complexity of phrase recognition, but also achieve efficient phrase search and result verification.

2) We propose a method that enables reliable verification of phrase search results. In our scheme, the verification evidence calculation and verification process of phrase search are both executed by the blockchain, breaking the pattern of the server generating both search results and verification evidence, so the reliability of phrase search is ensured. Furthermore, we use a multiset hash function to calculate cumulative evidence, which significantly reduces the overhead of the blockchain.

3) We conducted a security analysis of the scheme and conducted detailed experiments. The results demonstrate that our construction is secure and enjoys good search efficiency.

The article is structured as follows: "Related Work" introduces the current research progress related to phrase search and verification; "Problem Formulation" describes the system model, threat model, algorithm definitions, and security definitions; "Methods" provides a detailed description of the phrase search and verification algorithms used in our scheme; and finally, "Security Analysis" and "Results" respectively analyze the security and experimental results of the proposed solution.

## RELATED WORK

Searchable symmetric encryption (SSE) was first proposed by *Song, Wagner & Perrig (2000)* in 2000, which provides users with a new way to perform retrieval on encrypted data. However, this scheme uses full-text matching, and the search time is linear. To improve the search efficiency, *Anand et al. (2014)* proposed an efficient searchable encryption scheme with the inverted index, achieving a subcaptionlinear search. Following this direction, a great many schemes have been proposed to support dynamic update (*Kamara, Papamanthou & Roeder, 2012*; *Stefanov, Papamanthou & Shi, 2014*; *Liu et al., 2021*), multi-client query (*Sun, Zuo & Liu, 2022*; *Du et al., 2020*) and privacy protection (*Liu et al., 2014*; *Song et al., 2021*). But these schemes are mainly focusing on a single keyword, and the cloud returns some irrelevant files. To further improve the search efficiency and accuracy, SSE schemes supporting multi-keyword search are proposed, such as boolean query (*Cash et al., 2013*) and conjunctive queries (*Lai et al., 2018*). Compared with single keyword query, multi-keyword search improves search accuracy and reduces the communication and storage overhead.

Phrase search is a special case of multi-keyword search, it requires a sequential relationship between multiple keywords. *Anand et al. (2014)* first defined the model of phrase search and its security definition, but it is impractical in real scenarios since the client and the server require two rounds of interaction to complete a phrase query. *Poon & Miri (2015)* proposed a low storage phrase search scheme using bloom filter and symmetric encryption, and further proposed a fast phrase search scheme based on n-gram filters in 2019 (*Poon & Miri, 2019*). *Li et al. (2015)* implemented phrase search based on relative position, and realizes lightweight transactions and storage during the retrieval process. *Ge et al. (2021)* proposed an intelligent fuzzy phrase search scheme over encrypted network data for IoT, which dentifies phrases through binary matrices and look-up tables, and uses a fuzzy keyword set to resolve spelling errors in phrase searches. *Shen et al. (2019)* proposed a phrase search scheme that protects user privacy, which uses homomorphic encryption and bilinear mapping to achieve phrase identification.

Verifiable search: As we all know, servers in SSE are not completely trusted and may return incorrect search results due to external or internal attacks, so verifiable search is necessary. The concept of verifiable searchable symmetric encryption (VSSE) was first proposed by *Qi & Gong (2012)* in 2012, since then, a series of VSSE schemes are proposed (*Liu et al., 2016*; *Miao et al., 2021*; *Chen et al., 2021*; *Wu et al., 2023*). Unfortunately, these schemes are valid for a single keyword but do not support multiple keywords. *Wan & Deng (2018)* used homomorphic MAC to design a scheme that can verify the search results of multiple keywords. *Li et al. (2021)* used RSA accumulators to verify multi-keyword search results and uses bitmaps to improve search efficiency. There are similar multi-keyword verifiable ciphertext retrieval schemes (*Liu et al., 2021*; *Liang et al., 2020*, *2021*). *Kissel & Wang (2013)* utilized a validation tag to build a verifiable phrase search scheme over encrypted data, but they failed to verify the integrity of the file. For more complex phrase searches, *Ge et al. (2021)* used the MAC function and look-up tables to implement phrase search result verification. Although this construction can verify the phrase search, it adopts a two-phase query strategy, which means the user needs to interact with the server twice in a phrase search and generate a large number of trapdoors.

Verifiable search based on blockchain: In the above verifiable schemes, the server sends the search results and verification evidence to the user, and the user calculates the search results and compares them with the received evidence to complete the verification. But this approach has some disadvantages. First, the results and evidence are unreliable due to the server is untrusted. Second,this approach cannot solve the problem of fair verification between server and user. To address this problem, blockchain is introduced into verifiable search. Currently, some verifiable search solutions based on blockchain have been proposed (*Hu et al., 2018*; *Li et al., 2019*; *Guo, Zhang & Jia, 2020*), but these solutions mainly focus on single keyword search, while there are almost no reliable and fair verification solutions for multi-keyword search scenarios. The same is true for phrase searches, which are more complex than multi-keyword searches.

## PROBLEM FORMULATION

In this section, we formally define the efficient and reliable phrase search scheme over encrypted cloud-IoT data. We present the system model, threat model and security definition. We denote a composite index as a *secure index*, a searched phrase as a *query* and an encrypted query as a *trapdoor*. The notations and symbols used in our system are shown in Table 1.

### System model

Four entities are included in our system: IoT device, data user, cloud server, and blockchain. The system model is shown in Fig. 1. IoT device as the data owner collects data and stores them in the form of files $\mathscr{F} = \{F_1, F_2, \ldots F_M\}$. The IoT device extracts all the keywords $\mathscr{W} = \{w_1, w_2, \ldots w_N\}$ in $\mathscr{F}$ and adopts the bitmap to build composite index $\mathscr{I}$. The IoT device encrypts all the files in $\mathscr{F}$ to ciphertexts $\mathscr{C} = \{C_1, C_2, \ldots C_M\}$ by symmetric encryption, and calculates the hash value $hash_i$ of each ciphertext in $\mathscr{C}$ through

**Table 1 Notations and symbols.**

| Notation | Definition |
|---|---|
| $id_i$ | The identifier of the file $F_i$ |
| $M$ | The number of files |
| $N$ | The number of keywords |
| $\mathscr{L}(\cdot)$ | A bit-length of $\cdot$ |
| $|\cdot|$ | Number of elements in set $\cdot$ |
| $SL^i_{w_1}\big|_{|A|}$ | Get the first $|A|$ bits of $SL^i_{w_1}$ |
| $SL^i_{w_1}\big|_{-(|A|-|B|)}$ | Get the last $(|A|-|B|)$ bits of $SL^i_{w_1}$ |
| $\widetilde{w}$ | The query phrase |
| $R$ | A set of ciphertext satisfying phrase search |
| $proof$ | Verification result, 1: valid, 0: invalid |
| $\|$ | Concatenation symbol, $a\|b$ denotes the concatenation of message a and b. |
| $r_{i,j}$ | Number of positions of keyword $w_i$ in file $F_j$ |

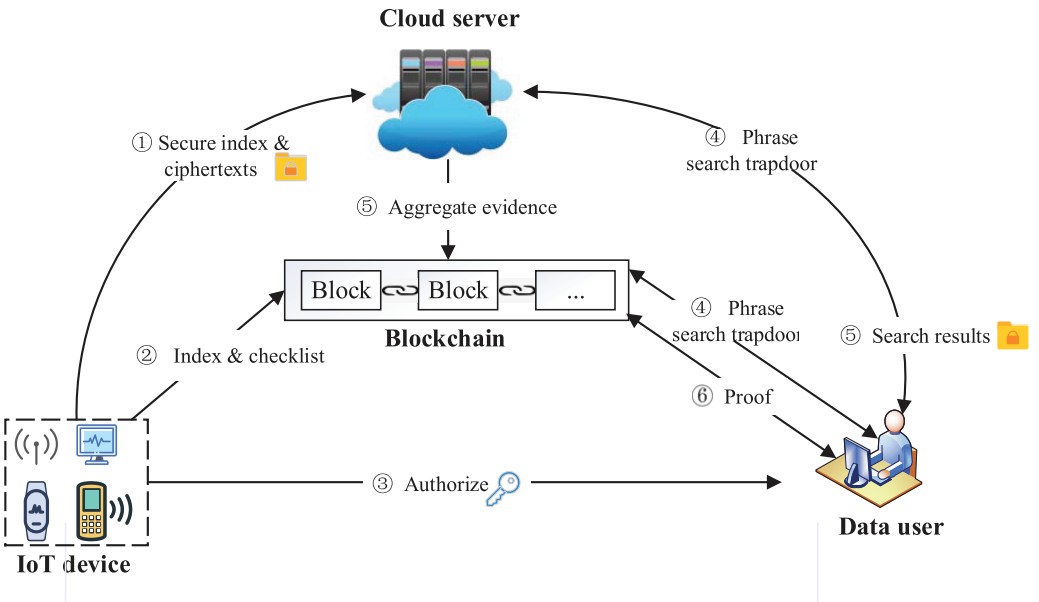

**Figure 1 System model.** Image credit: component source from https://www.iconfont.cn/.

sha256, which will be added to the checklist $L$. At last, $(\mathscr{I}, \mathscr{C})$ and $(\mathscr{I}, L)$ are sent to the cloud server and the blockchain, respectively.

The data user obtains the system public parameters $\Omega$ through the authorization of the IoT device, generates a search trapdoor through these public parameters and the phrases to be queried $\widetilde{w}$. The trapdoor will be sent to the server and blockchain for encrypted search and result verification, respectively. The data user receives the search results $R$ and verification result $proof$ from the blockchain, and accepts $R$ if $proof = 1$, otherwise rejects $R$.

The cloud server stores the index $\mathscr{I}$ and ciphertexts $\mathscr{C}$ sent by the IoT device, and performs a search over encrypted data using the trapdoor sent by the data user to generate the search result $R$ and aggregate evidence $\psi$. At last, the cloud server sends $\psi$ to the blockchain for verification, and sends $R$ to the user.

The blockchain verifies aggregate evidence $\psi$ returned by the server and generates *proof*. To achieve reliable verification, the blockchain performs a phrase search in parallel with the cloud server to generate the verification standard value $\psi'$. The blockchain compares $\psi'$ with the aggregated evidence $\psi$ returned by server, calculates the verification evidence *proof*, and sends it to the user. In particular, during the verification, multi-set hash functions are used to verify the aggregate hash results of the ciphertext, while the ciphertext is off-chain, thereby reducing blockchain storage and computing overhead.

## Threat model

In our system, IoT devices and blockchains are completely trusted, IoT devices can collect data honestly, and generate secure indexes and checklists. The blockchain performs fair verification of search results, and the verification result is reliable and unforgeable.

Correspondingly, cloud servers and users are considered untrustworthy. The cloud server may only store part of the index and ciphertext for saving storage resources. At the same time, it may perform searches dishonestly in order to save computation costs. In addition, there may be other software/hardware malfunctions in the system. All the above reasons will make the file and the verification evidence returned by the server incomplete or incorrect. As for data users, it may falsify verification results for financial gain and is therefore not trustworthy.

## Algorithm definitions

Our scheme consists of six polynomial algorithms
$\prod = \{KeyGen, IndexBuild, TokenGen, Search, Verify, Dec\}$:

1) $K \leftarrow KeyGen(1^\lambda)$, this algorithm inputs a secure parameter $\lambda$, and outputs the key set $K = (K_1, K_2, K_3, K_4, K_I, K_Z, K_X, pk, sk)$.

2) $(\mathscr{I}, T, B) \leftarrow IndexBuild(\mathscr{F}, \mathscr{W}, K)$, this algorithm takes the set of files $\mathscr{F}$, the set of keywords $\mathscr{W}$, the key set $K$ as input, and outputs the secure index $\mathscr{I}$, the encrypted database $T$, the checklist $B$.

3) $TK_{i,Q} \leftarrow TokenGen(\widetilde{w}, K_3, pk)$, this algorithm takes a query phrase $\widetilde{w}$, a secret key $K_3$ and a public key $pk$ as input, and outputs the search trapdoor $TK_{i,Q}$.

4) $(\psi, R) \leftarrow Search(\mathscr{I}, T, TK_{i,Q})$, this algorithm takes the secure index $\mathscr{I}$, the encrypted database $T$ and the search trapdoor $TK_{i,Q}$ as input, and outputs the aggregate evidence $\psi$ and search results $R$.

5) $proof \leftarrow Verify(\mathscr{I}, \psi, B, TK_{i,Q})$, this algorithm takes the secure index $\mathscr{I}$, aggregate evidence $\psi$, the checklist $B$, the search trapdoor $TK_{i,Q}$ as input, and outputs the verification evidence *proof*.

6) $F \leftarrow Dec(K_2, C)$, this algorithm takes the symmetric key $K_2$ and the encrypted file $C$ as input, and outputs the plaintext $F$.

## Leakage function

The goal of searchable encryption is to leak as little information as possible about the keywords and files during ciphertext retrieval. Similar to *Wu et al. (2023)*, the leak function is defined as $\mathscr{L} = \{\mathscr{L}_{IndexBuild}, \mathscr{L}_{Search}, \mathscr{L}_{Verify}\}$. According to the common definition, query history $Hist = \{(DB_i, q_i)\}_{i=0}^{n}$, which stores a series of query requests and corresponding database snapshots. The search pattern $sp(w) = \{i|$for each $q_i$ that contains $w$ in **Hist**$\}$, which records each query request. The proof history $ph(w) = \{(i, proof_i)$ $|$ *for each* $(i, ind_i, w)$ *in Hist*$\}$. Then, we can define the leakage function $\mathscr{L}_{IndexBuild} = (ph(w))$, $\mathscr{L}_{Search} = (sp(w), ph(w))$ and $\mathscr{L}_{Verify} = (ph(w))$.

## Security definitions

Definition 1 (Verifiability). In an efficient and verifiable phrase search scheme, if the probability that the forged result generated by any probabilistic polynomial time (PPT) adversary passes the Verify algorithm is infinitesimal, the scheme satisfies verifiability.

Definition 2 (CKA2-security). For the verifiable phase search scheme $\prod =$ $\{KeyGen, IndexBuild, TokenGen, Search, Verify, Dec\}$, there is a leakage function $\mathscr{L} = \{\mathscr{L}_{IndexBuild}, \mathscr{L}_{Search}, \mathscr{L}_{Verify}\}$, an adversary $\mathscr{A}$ and an idealized simulator $\mathscr{S}$, as well as two games $\text{Real}_{\mathscr{A}}(\lambda)$ and $\text{Ideal}_{\mathscr{A},\mathscr{S}}(\lambda)$, satisfying:

$\text{Real}_{\mathscr{A}}(\lambda)$: The challenger generates system key $K = \{K_1, K_2, K_3\}$ and index $(\mathscr{I}, T, B)$ by executing algorithm $KeyGen(1^\lambda)$ and algorithm $IndexBuild(\mathscr{F}, \mathscr{W}, K), (\mathscr{I}, T, B)$ are transmitted to the adversary $\mathscr{A}$. $\mathscr{A}$ proceeds to formulate a sequence of adaptive queries $Q = \{q_1, q_2, \ldots, q_t\}$, with the challenger generating search tokens for each query, and receives the results of executing algorithms $Search$ and $Verify$. Finally, $\mathscr{A}$ produces a bit $b$ as the output of this experiment.

$\text{Ideal}_{\mathscr{A},\mathscr{S}}(\lambda)$: The simulator $\mathscr{S}$ takes $(F, W)$ generated by the adversary $\mathscr{A}$ as input and outputs index $(\mathscr{I}, T, B)$ by executing algorithm $\mathscr{L}_{IndexBuild}$. Then, for a series of adaptive queries $Q = \{q_1, q_2, \ldots, q_t\}$ generated by the adversary $\mathscr{A}$, $\mathscr{S}$ generates search results by executing algorithms $\mathscr{L}_{Search}$ and $\mathscr{L}_{Verify}$, $\mathscr{A}$ receives those results and produces a bit $b$ as the output of this experiment.

If there is a simulator $\mathscr{S}$ such that for any PPT adversary $\mathscr{A}$:

$$| \Pr[\text{Real}_{\mathscr{A}}(\lambda) = 1] - \Pr[\text{Ideal}_{\mathscr{A},\mathscr{S}}(\lambda)] = \leq negl(\lambda),$$

then $\prod$ is $\mathscr{L}$–secure against adaptive chosen-keyword attack (CKA2), where *negl* is an negligible function and $\lambda$ is the security parameter.

## Preliminaries

Bitmaps employ binary strings to represent information sets, commonly utilized for storing file identifiers in encrypted searches, thus efficiently reducing storage requirements. In our model, each keyword $w_i$ corresponds to a bitmap, and the bitmap is a string composed of a series of 0 or 1, each 0 or 1 denotes a file. If the $i - th$ document contains $w_i$, the value of the string at position $i$ is set to 1, otherwise 0. For instance, with four files ($F_1$, $F_2$, $F_3$, $F_4$) and two keywords ($w_1$, $w_2$) in the system, depicted in Fig. 2, $w_1$ is found in $F_1$

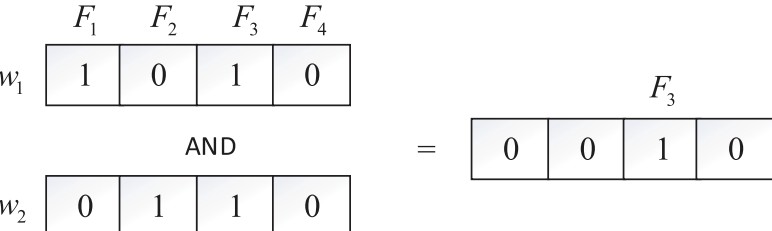

**Figure 2 Bitmap.**

and $F_3$, while $w_2$ exists in $F_2$ and $F_3$. The bitmaps for $w_1$ and $w_2$ are 1010 and 0110, respectively. To search for files containing both $w_1$ and $w_2$, an "AND" operation on these two bitmaps is performed, yielding $1010 \wedge 0110 = 0010$, indicating that $F_3$ contains both $w_1$ and $w_2$.

Homomorphic encryption represents an encryption technique capable of transforming a ciphertext into another without altering the decryption key. In this study, we employ the prevalent Paillier additive homomorphic encryption to compute the distance between keywords within phrases. In essence, its functionality can be outlined as follows:

1) Key Generation: Let $p$ and $q$ denote two large primes such that $\gcd(pq, (p-1) \times (q-1)) = 1$. Define $n = pq$ and $\lambda = lcm(p-1, q-1)$. Choose a random integer $g$ from $Z_{n^2}^*$ satisfying $\gcd(L(g^\lambda \bmod n^2), n) = 1$, where $L(x) = (x-1)/n$ and $Z_{n^2}^* = \{1, 2, \ldots, n^2 - 1\}$. Then, the public key $(n, g)$ and the private key $\lambda$ are obtained.

2) Encryption: Given a message $m$, it can be encrypted into its ciphertext $c$ as follows:

$$c = E(m, r) = g^m r^n \bmod n^2 \tag{1}$$

where $r$ is randomly selected from $r \in Z_n$.

3) *Decryption*: For the ciphertext $c$, it can be decrypted into its plaintext $m$ as follows:

$$m = D(c, \lambda) = \frac{L(c^\lambda \bmod n^2)}{L(g^\lambda \bmod n^2)} \bmod n \tag{2}$$

This algorithm exhibits additive homomorphism. Given two messages $a$ and $b$ along with their corresponding ciphertexts $E(a)$ and $E(b)$, we can obtain the ciphertext of $(a + b)$ via $E(a) \cdot E(b)$, i.e., $E(a + b) = E(a) \cdot E(b)$. This property can be leveraged to compute the distance between keywords in a phrase, aiding in determining their positional relationship.

Multiset Hash Function (*Li et al., 2023*): Multiset hash is a cryptographic tool that maps multiple sets of any finite size to a fixed hash length. Furthermore, multiset hash is also updateable: when the elements in the set change, the hash value only updates the current value without recalculating all.

Our scheme uses the most efficient multi-set hash function: MSet-XOR-Hash, containing three polynomial algorithms $(\mathscr{H}, +_{\mathscr{H}}, \equiv_{\mathscr{H}})$. Given a multiset $M$, the MSet-XOR-Hash can be expressed as follows:

$$\begin{cases} \mathcal{H}(r, M) & = H_k(0, r) \oplus \bigoplus_{m \in M} H_k(1, m); \\ \mathcal{H}(r, M \cup \{x\}) & \equiv {}_{\mathcal{H}}\mathcal{H}(r, M) + {}_{\mathcal{H}}\mathcal{H}(r, \{x\}) \\ & \equiv {}_{\mathcal{H}}\mathcal{H}(r, M) \oplus H_k(1, x); \\ \mathcal{H}(r, M \setminus \{x\}) & \equiv {}_{\mathcal{H}}\mathcal{H}(r, M) - {}_{\mathcal{H}}\mathcal{H}(r, \{x\}) \\ & \equiv {}_{\mathcal{H}}\mathcal{H}(r, M) \oplus H_k(1, x) \end{cases}$$

## METHODS

We present the construction of the efficient and reliable phrase search scheme over encrypted cloud-IoT data in this section.

### Composite index containing files and locations

In phrase search, a phrase is composed of multiple keywords according to a certain positional relationship, which is also the difference between phrase search and multi-keyword search. To perform a phrase search, the cloud server not only needs to search for all keywords contained in the phrase, but also needs to determine whether the order between keywords is correct.

To identify the position relationship between keywords in phrases, we designed a composite index containing files and locations, the structure of the composite index is shown in Fig. 3.

The composite index adopts inverted index structure to ensure high efficiency of search, but, it's different from the general inverted index in that each keyword not only corresponds to the ID of a series of files, but also appends all the locations where the keyword appears in the file. For example, in Fig. 3, suppose there are three keywords ("*heart*", "*attack*", "*medic*") and five corresponding files ($F_1, F_2, F_3, F_4, F_5$), for simplicity, encryptions are not shown. The positions of keyword "*heart*" in files $F_1, F_2, F_3$ are (1, 8, 3), (1, 2, 4) and (2, 3, 5) respectively, the positions of keyword "*attack*" in files $F_1, F_2, F_4$ are (2, 5, 7), (3, 7, 9) and (1, 2, 5). When the cloud server searches the phrase "*heart attack*", it finds that the location of keyword "*heart*" in $F_1$ is (1, 8, 3) through the composite index, then it finds the location of keyword "*attack*" in $F_1$ is (2, 5, 7). Using the encrypted distance discrimination algorithm, the cloud server computes the position of "*attack*" in file $F_1$ is 1 larger than that of "*heart*" by $E(2) = E(1) \cdot E(1)$. Similarly, the cloud server computes that "*attack*" is after "*heart*" in $F_2$ through $E(3) = E(2) \cdot E(1)$. After searching the location of all keywords in the composite index, the server calculates that $F_1$ and $F_2$ contain the phrase "*heart attack*".

### Encrypted distance discrimination algorithm-EDDA

The sequence of keywords in a phrase can be expressed by a sentinel and the distance between each remaining keyword and the sentinel. For example, in a phrase containing three keywords ($w_1, w_2, w_3$), the position of $w_1, w_2, w_3$ are 1, 2, 3, we choose $w_1$ as the sentinel. The distance between $w_2$ and $w_1$ is 1, and the distance between $w_3$ and $w_1$ is 2. Suppose that positions of ($w_1, w_2, w_3$) are ($pos_1, pos_2, pos_3$), if we can

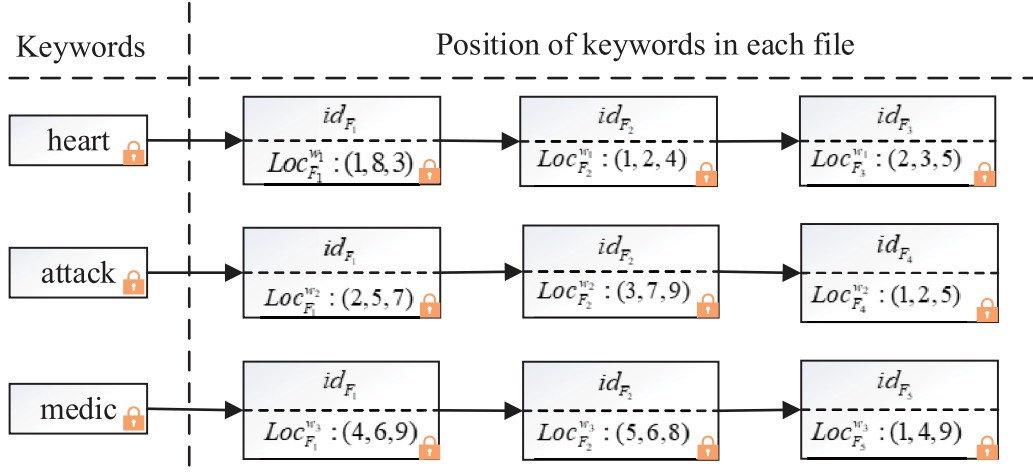

**Figure 3  Example of the composite index structure.**

calculate $pos_2 = pos_1 + 1$ and $pos_3 = pos_1 + 2$, we can recognize that $(w_1, w_2, w_3)$ is a phrase.

In our scheme, the positions of keywords stored in the composite index are encrypted, and the server should be able to recognize phrases without the decryption key. Therefore, we utilize the paillier homomorphic encryption to construct the distance discrimination algorithm to determine the location relationship between keywords, the details are as follows:

$E(d)$ is the distance after paillier homomorphic encryption, and $SL^i_{w_j}$ denotes the encrypted location of the keyword $w_j$ in file $id_i$, the definition of $SL^i_{w_j}$ is as follows:

$$SL^i_{w_j} = \pi(id_i) + E(pos^i_{w_j}) \tag{3}$$

$E$ represents paillier homomorphic encryption function and $\pi$ represents hash function, $pos^i_{w_j}$ is the original location of the keyword $w_j$ in file $id_i$. In addition, keyword $w_j$ may appears in multiple locations in the same file, in this case, $E(pos^i_{w_j})$ represents a series of positions.

In distance discrimination algorithm, $SL^i_{w_1}|_{|\pi|} \oplus SL^i_{w_2}|_{|\pi|}$ is used to determine whether $w_1$ and $w_2$ belong to the same file, if so, $val_i == 0$. $E(pos^i_{w_1}) \leftarrow SL^i_{w_j}|_{-(|SL^i_{w_j}| - |\pi|)}$ is used to calculate encrypted location of keyword $w_j$, and $E(pos^i_{w_2}) == E(pos^i_{w_1}) + E(d)$ is used to determine whether the keyword $w_2$ is located in the $d$ position after $w_1$. When the user executes the phrase search request, he can designate the first word (*i.e.*, $w_1$) in the phrase as the sentry, then calculate the distance $d$ between the remaining words and the sentry one by one, encrypt $d$, and finally generate a search token and send it to the server for search.

## Details of our construction

Like most searchable encryption schemes, we adopt an inverted index structure to construct the secure index. In the inverted index, we use a bitmap to store the identifier of

**Algorithm 1** Distance discrimination algorithm.

Input: $SL_{w_1}^i, SL_{w_2}^i, E(d)$

Output: Flag

1: $Flag \leftarrow False$

2: $val_i \& = SL_{w_1}^i|_{|\pi|} \oplus SL_{w_2}^i|_{|\pi|}$

3: if $val_i == 0$ then

4:      $E(pos_{w_1}^i) \leftarrow SL_{w_1}^i|_{-(|SL_{w_1}^i|-|\pi|)}$

5:      $E(pos_{w_2}^i) \leftarrow SL_{w_2}^i|_{-(|SL_{w_2}^i|-|\pi|)}$

6:      if $E(pos_{w_2}^i) == E(pos_{w_1}^i) + E(d)$ then

7:         $Flag \leftarrow True$

8:      end if

9: end if

10: return Flag

the file. Let $H : \{0,1\}^* \rightarrow \{0,1\}^m, F : \{0,1\}^* \rightarrow \{0,1\}^n$ be secure pseudo-random functions (PRFs).

*KeyGen*$(1^\lambda)$. The IoT device uses the secret parameters $\lambda$ to generate the key set $K = \{K_1, K_2, K_3, K_4, K_I, K_Z, K_X, pk, sk\}$, where $K_1, K_2, K_3, K_4, K_I, K_Z, K_X \xleftarrow{\$} \{0,1\}^\lambda$, $(pk, sk) \leftarrow Paillier.KeyGen(1^\lambda)$. $K_1$ is used to encrypt the identifier of files, $K_2$ is the secret key of symmetric encryption, $K_3$ is the key for PRF $F$, $(pk, sk)$ is the public key and the private key of paillier encryption.

*IndexBuild*$(\mathcal{F}, \mathcal{W}, K)$. Given a set of files $\mathcal{F}$, a set of keywords $\mathcal{W}$, and the key set $K$, the IoT device generates the secure index $\mathcal{I}$, the encrypted database $T$, the checklist $B$, the details are shown in Algorithm 2.

For each file $F_i \in \mathcal{F}$, IoT device encrypts it to the ciphertext $C_i$ with symmetric encryption $Enc(K_2, F_i)$. The ciphertext $C_i$ is stored in encrypted database $T$, and the hash value $hash_i$ of $C_i$ is stored in checklist $B$ for verification.

IoT device generates a bitmap $\mathcal{B}_{w_j}$ for each keyword $w_j$, $\mathcal{B}_{w_j}$ is encrypted by $v_{\mathcal{B}} \leftarrow \mathcal{B}_{w_j} \oplus H_2(u_{w_j}||st_j)$ and $v_{\mathcal{B}}$ is stored in the secure index $\mathcal{I}$. Especially, in order to protect the privacy of files, IoT device uses $\ell_i \leftarrow H_2(id_i||K_1)$ to encrypt the *id* of files, and then uses $\ell_i$ to generate $\mathcal{B}_{w_j}$. Since the *id* stored on the server is encrypted, the server cannot obtain the real *id* from $\mathcal{B}_{w_j}$, which ensures the privacy of the search pattern.

To identify phrases, the IoT device extracts positions $(pos_1, pos_2, \ldots, pos_m)$ of keyword $w_j$ in file $F_i$, and encrypts them using *Paillier.Enc*$(pos_m)$:

$$SL_{w_j}^i = \pi(id_i) + E(pos_1) + E(pos_2) + \ldots + E(pos_m)$$

*TokenGen*$(\widetilde{w}, K_3, pk)$. Authorized users get shared parameters $\Omega = \{K_3, pk\}$ from the IoT device. For the phrase $\widetilde{w} = \{w_1, w_2, \ldots, w_t\}$ to be queried, the trapdoor $TK_{i,Q}$ is generated as Algorithm 3.

---

**Algorithm 2  IndexBuild.**

Input: $DB, \mathscr{W}, K$

Output: $\mathscr{I}, T, B$

1: for $F_i \in \mathscr{F}$ do

2:   $\ell_i \leftarrow H(id_i||K_1); C_i \leftarrow Enc(K_2, F_i)$

3:   $hash_i \leftarrow H(C_i); B[\ell_i] \leftarrow hash_i; T[\ell_i] \leftarrow C_i$

4: end for

5: for $w_j \in \mathscr{W}$ do

6:   for $id \in DB$ do

7:     $u_{w_j} \leftarrow F_1(K_3, w_j); st_j \leftarrow F_2(K_4, id); t_{w_j} \leftarrow H_2(u_{w_j}||st_j)$

8:     Extract positions $(pos_1, pos_2, ..., pos_m)$ of keyword $w_j$ in file $F_i$

9:     $SL^i_{w_j} = \pi(id_i) + E(pos_1) + E(pos_2) + ... + E(pos_m)$

10:     $v_{\mathscr{B}} \leftarrow (\mathscr{B}_{w_j}||SL^i_{w_j}) \oplus H_2(u_{w_j}||st_j); \mathscr{I}[t_{w_j}] \leftarrow v_{\mathscr{B}}; \sum[w_j] = st_j$

11:   end for

12: end for

13: send $(\mathscr{I}, B)$ to blockchain, send $(\mathscr{I}, T)$ to cloud server

---

**Algorithm 3  TokenGen.**

Input: The query phrase $\tilde{w}$, the key set $K$

Output: The serach trapdoor $TK_{i,Q}$

1:  Suppose that query phrase $\tilde{w} = \{w_1, w_2, ..., w_t\}$

2:  for $j = 1 \rightarrow t$ do

3:    $st_j \leftarrow \sum[w_j], u_{w_j} \leftarrow F_1(K_3, w_j)$

4:    $t_{w_j} \leftarrow H_2(u_{w_j}||st_j)$

5:    if $j > 1$ then

6:      $d = j - 1; E_d \leftarrow Paillier.Enc(d)$

7:    end if

8:  end for

9:  send $TK_{i,Q} = \{t_{w_1}, t_{w_2}, ..., t_{w_t}, E_1, E_2, ..., E_{t-1}\}$ to blockchain and cloud server

---

Assume that the keywords $\{w_1, w_2, \ldots, w_t\}$ in phrase are arranged in order. The data user calculates the distance $d$ between the keyword $w_t$ and the first keyword $w_1$ and encrypts it with $Paillier.Enc(d)$. At last, $t_{w_j}$ and $E_d$ are added to the trapdoor $TK_{i,Q}$ and sent to the cloud server and the blockchain.

$Search(\mathscr{I}, T, TK_{i,Q})$. As shown in Algorithm 4, the cloud server performs an encrypted search with the secure index $\mathscr{I}$, the encrypted database $T$ and the trapdoor $TK_{i,Q}$.

After receiving the query request, the server parses the trapdoor $\{t_{w_1}, t_{w_2}, \ldots, t_{w_t}, E_1, E_2, ..E_{t-1}\} \leftarrow TK_{i,Q}$. The server gets the bitmap $\mathscr{B}_{w_i}$ of $w_i$ from the

---

**Algorithm 4** Search.

Input: $\mathscr{I}, T, TK_{i,Q}$

Output: Search results $R$

1: Parse $\{t_{w_1}, t_{w_2}, ..., t_{w_t}, E_1, E_2, ..E_{t-1}\} \leftarrow TK_{i,Q}$

2: **for** $t_{w_j} \in TK_{j,Q}$ **do**

3:      $v_{\mathscr{B}} \leftarrow \mathscr{I}[t_{w_j}]; \mathscr{B}_{w_j} \| SL^i_{w_j} \leftarrow v_{\mathscr{B}} \oplus H_2(t_{w_j})$

4: **end for**

5: $\mathscr{B} = \mathscr{B}_1 \wedge \mathscr{B}_2 \wedge ... \wedge \mathscr{B}_t, r = \mathscr{H}(\mathscr{B}, \{\perp\})$

6: Get $ID_{\mathscr{B}} = \{\ell_1, \ell_2, ..., \ell_p\}$ from $\mathscr{B}$

7: **for** $\ell_i \in ID_{\mathscr{B}}$ **do**

8:      $flag = \{000...000\}^{t-1}, hash_i = H_1(t[\ell_i]), \psi = \psi +_{\mathscr{H}} \mathscr{H}(r, hash_i)$

9:      $(E(pos^1_1), E(pos^1_2), E(pos^1_m)) \leftarrow SL^1_{w_1}$

10:      **for** $d = 2 \rightarrow t$ **do**

11:          $(E(pos^i_1), E(pos^i_2), E(pos^i_n)) \leftarrow SL^i_{w_j}$

12:          **for** $k = 1 \rightarrow m; k' = 1 \rightarrow n$ **do**

13:              **if** $E(pos^{k'}_1) = E(pos^k_1) \times E(d-1)$ **then**

14:                  Set the position $(i-1)$ of $flag$ to 1

15:                  **break**

16:              **end if**

17:          **end for**

18:      **end for**

19:      If all positions of $flag$ are 1

20:      get $C_i \leftarrow T[\ell_i], R \leftarrow R \cup C_i$

21: **end for**

22: Server sends $\{\psi\}$ to the blockchain for verification, and sends $R$ to the data user

---

secure index $\mathscr{I}$ through $v_{\mathscr{B}} \leftarrow \mathscr{I}[t_{w_i}], \mathscr{B}_{w_i} \leftarrow v_{\mathscr{B}} \oplus H(t_{w_i})$. To get the file that contains all the keywords in the phrase $\tilde{w}$, the server performs the operation "*AND*" on the bitmap of all keywords as follows:

$$\mathscr{B} = \mathscr{B}_1 \wedge \mathscr{B}_2 \wedge \ldots \wedge \mathscr{B}_t.$$

The file corresponding to the element with a value of "1" in $\mathscr{B}$ contains all the keywords in the phrase. The server gets the set of identifiers of these files as $ID_{\mathscr{B}} = \{\ell_1, \ell_2, \ldots, \ell_p\}$ according to $\mathscr{B}$.

Next, the server determines whether the sequence of the keywords in the file $\ell_i$ is consistent with the order of the keywords in the phrase, as described in line 7–line 20 in Algorithm 3. The server chooses a binary string $flag$ of length $(t-1)$, and set all values to "0". The server gets all positions $E(pos^i_1), E(pos^i_2), E(pos^i_n)$ of the keyword $w_j$ in the file $\ell_i$. For the position $E(pos^{k'}_1)$ of the keyword $w_j (j > 1)$ in file $\ell_i$, the server utilizes the distance

discrimination algorithm *EDDA* to determine the distance between keyword $w_j(j > 1)$ and the first keyword $w_1$ in the phrase. Like

$$E(pos_1^{k'}) = E(pos_1^k) \times E(d - 1) \tag{4}$$

where *E* represents the *Paillier.Enc*. If Formula (4) holds, the distance between keywords $w_1$ and $w_j$ is $(d - 1)$, which is the same as that in the phrase, the server sets the position $(i - 1)$ of *flag* to "1". If all positions of *flag* are "1", then the file $\ell_i$ contains the phrase $\widetilde{w}$, and it is added to the search result *R*. Finally, the aggregation proof $\psi$ is sent to the blockchain for reliable verification, and the ciphertext collection of search results *R* is sent to the user.

*Verify*($\mathscr{I}, B, TK_{i,Q}, \psi$). The blockchain utilizes $TK_{i,Q}$ for phrase searches to verify the aggregated evidence $\psi$ returned by the cloud server, as shown in Algorithm 5. To ensure the reliable verification of search results, the verification algorithm *Verify* not only verifies the integrity of the files, but also verifies whether the server has returned all files that meet the search requirements.

The blockchain performs the same operations as the server (line 1–line 6), retrieving the composite index stored on itself with the trapdoor $TK_{i,Q}$, and calculates the search result $\psi'$. Due to the immutability of data on the blockchain, the composite index stored and search results calculated by the blockchain are reliable. Blockchain achieves reliable verification of phrase search results by comparing $\psi'$ with the search result $\psi$ returned by the server.

For the file $\ell_i$, the blockchain obtains the corresponding hash value $hash_i$ by searching the checklist *B* and compresses it into the benchmark value $\psi'$. By comparing the aggregate evidence $\psi$ sent by the server with $\psi'$, the blockchain sets the value of *proof* as follows:

$$proof = \begin{cases} 1, & \text{if } \psi = \psi', \\ 0, & \text{otherwise.} \end{cases}$$

By comparing $\psi = \psi'$, the blockchain can determine: 1) whether the server has returned all files that meet the search requirements; 2) the content of the files has been modified.

Then the verification evidence *proof* are sent to the data user. The data user judges the received *proof*, and accepts the search result *R* if *proof* = 1, otherwise rejects *R*. For the accepted search result *R*, the data user uses the symmetric key to decrypt the file in it, to get the plaintext of the file, and the phrase search process is completed.

## Discussion

Ensuring the reliability of verification is an important target of our scheme. In the existing phrase search scheme, the secure index is stored on the server, and the verification evidence is generated by the server. Untrusted servers may only store partial indexes and ciphertexts, resulting in untrustworthy search results and verification evidence. Whereas in our scheme, blockchain uses search trapdoor to calculate verification evidence, the data stored on the blockchain is unforgeable, so the search results on the blockchain are reliable. At the same time, the verification of the results returned by the server is also performed by

---

**Algorithm 5   Verify.**

Input: $\mathscr{I}, B, TK_{i,Q}, \psi$

Output: *proof*

1:  $\psi' \leftarrow \phi, proof = 0.$

2:  Using search trapdoors $TK_{i,Q}$ to perform phrase searches same as line 1–line 6 of Algorithm 4.

3:  for $\ell_i \in ID_{\mathscr{B}}$ do

4:      $hash_i' \leftarrow B[\ell_i], \psi' \leftarrow +_{\mathscr{H}} \mathscr{H}(r, hash_i)$

5:  end for

6:  if $\psi = \psi'$ then

7:      $proof = 1$

8:  end if

9:  The blockchain sends *proof* to the data user.

---

the blockchain, which prevents dishonest data users from falsifying the verification results and ensures the reliability of the verification results.

## SECURITY ANALYSIS

Theorem 1: The proposed efficient and reliable phrase search scheme satisfies verifiability.

Proof. Let $\mathscr{A}$ be a PPT adversary who can produce a forgery $R_{\mathscr{S}}$, which can pass the verification algorithm *Verify*. Assuming the correct search result is $R$, we will prove that there is no such adversary $\mathscr{A}$ who can give a forgery $R_{\mathscr{S}}$ satisfying $R = R_{\mathscr{S}}$.

Suppose the compressed hash values corresponding to $R$ and $R_{\mathscr{S}}$ are $\psi$ and $\psi_{\mathscr{S}}$, respectively, and we will discuss the following two cases:

Case 1: $R = R_{\mathscr{S}}$ and $\psi \neq \psi_{\mathscr{S}}$. For each ciphertext $C_j$ in $R$, we have $hash_j \leftarrow H(C_j), \psi = \psi +_{\mathscr{H}} \mathscr{H}(r, hash_j)$, similarly, we have $hash_j^{\mathscr{S}} \leftarrow H_{\mathscr{S}}(C_j^{\mathscr{S}}), \psi_{\mathscr{S}} = \psi_{\mathscr{S}} +_{\mathscr{H}} \mathscr{H}(r, hash_j^{\mathscr{S}})$ for each ciphertext $C_j^{\mathscr{S}}$ in $R_{\mathscr{S}}$. Since the data on the blockchain is unforgeable and $R = R_{\mathscr{S}}$, we have $\psi = \psi_{\mathscr{S}}$, which is contradictory to $\psi \neq \psi_{\mathscr{S}}$. Therefore this case does not hold.

Case 2: $R \neq R_{\mathscr{S}}$ and $\psi = \psi_{\mathscr{S}}$. This implies that $\mathscr{A}$ can discover a collision for $H$, which contradicts the collision resistance property of the hash function. Therefore, this case also does not hold.

In summary, the unforgeability of blockchain and the collision resistance of hash function ensures that any PPT adversary $\mathscr{A}$ cannot forge search results. So, our scheme satisfies verifiability.

Theorem 2: If PRF $F$ is pseudo-random, *Enc* algorithm is secure against chosen plaintext attack (CPA-secure) and *Paillier.Enc* is secure against chosen ciphertext attack (CCA-secure), then our proposed scheme is $(\mathscr{L}_{IndexBuild}, \mathscr{L}_{Search}, \mathscr{L}_{Verify})$-secure against the adaptive chosen-keyword attack.

Proof. We establish the CKA2 security of our scheme by demonstrating the indistinguishability of $\text{Real}_{\mathscr{A}}(\lambda)$ and $\text{Ideal}_{\mathscr{A},\mathscr{S}}(\lambda)$. The proof starts with $\text{Real}_{\mathscr{A}}(\lambda)$ and go

through a series of indistinguishable games to achieve $\text{Ideal}_{\mathscr{A},\mathscr{S}}(\lambda)$, thus proving that A and $\text{Ideal}_{\mathscr{A},\mathscr{S}}(\lambda)$ are indistinguishable.

Game $\mathbf{G_1}$: $G_1$ is the same with $\text{Real}_{\mathscr{A}}(\lambda)$:

$$|\Pr[\text{Real}_{\mathscr{A}}(\lambda) = 1] = \Pr[G_1 = 1]$$

Game $\mathbf{G_2}$: We replace the output of the pseudorandom function $F(F_1$ and $F_2)$ with a sequence of binary random numbers $\hat{F}$, the length of $\hat{F}$ is equal to $|F|$, and store the binary sequence in buckets $B_1$ and $B_2$. If the adversary $\mathscr{A}$ can distinguish between $F$ and the random number sequence, then they can distinguish between $\mathbf{G_1}$ and $\mathbf{G_2}$. Then,

$$|\Pr[G_1 = 1] - \Pr[G_0 = 1] \leq Adv_{F_1,F_2,\mathscr{A}}^{PRF}(\lambda)$$

Game $\mathbf{G_3}$: In $G_3$, the output of the hash function $H(H_1$ and $H_2)$ is replaced by a series of randomly generated binary strings $\hat{H}$, $|\hat{H}| = |H|$. $G_3$ stores $\hat{H}$ in buckets $HB_1$ and $HB_2$. If the adversary $\mathscr{A}$ can distinguish between $H$ and $\hat{H}$, then they can distinguish between $\mathbf{G_2}$ and $\mathbf{G_3}$. Then,

$$|\Pr[G_3 = 1] - \Pr[G_2 = 1] \leq negl(\lambda)$$

Game $\mathbf{G_4}$: In $G_3$, the output of the multi-set hash function is computed based on $(r, hashi)$, while in $G_4$, the output of the multi-set hash function consists of a random binary string made up of a series of 0 or 1. And, the binary string is recorded in a bucket $\hat{X}$. From the previous analysis, we can conclude that

$$|\Pr[G_4 = 1] - \Pr[G_3 = 1] \leq negl(\lambda)$$

$\text{Ideal}_{\mathscr{A},\mathscr{S}}(\lambda)$: $\text{Ideal}_{\mathscr{A},\mathscr{S}}(\lambda)$ and $G_4$ are the same, except that $\text{Ideal}_{\mathscr{A},\mathscr{S}}(\lambda)$ introduces simulator $\mathscr{S}$, $\mathscr{S}$ executes algorithm $\mathscr{L}_{IndexBuild}, \mathscr{L}_{Search}, \mathscr{L}_{Verify}$ with the help of $(sp(w), ph(w))$ and the adversary $\mathscr{A}$ can sniff the algorithm output. The algorithm details are shown in Algorithms 6–8. The adversary $\mathscr{A}$ cannot distinguish between the output of the random oracle in this game and the actual data, hence

$$|\Pr[\text{Ideal}_{\mathscr{A},\mathscr{S}}(\lambda) = 1] - \Pr[G_4 = 1] \leq negl(\lambda)$$

From what we have discussed above, the adversary cannot distinguish the result in the experiment **Real** and the result in the experiment **Ideal**. That is:

$$|\Pr[\text{Real}_{\mathscr{A}}(\lambda) = 1] - \Pr[\text{Ideal}_{\mathscr{A},\mathscr{S}}(\lambda) = 1] \leq negl(\lambda)$$

Therefore, our proposed scheme satisfies CKA2 security.

## RESULTS

In order to objectively evaluate the performance of our scheme, we design a series of scientific experiments in this section. We conducted a comprehensive analysis of the experimental results and compared them with the existing phrase search scheme (*Kissel & Wang, 2013*) and scheme (*Ge et al., 2021*). Our experiments are deployed on a local laptop

---

**Algorithm 6**  **Simulator 6.**

Input: $DB, \mathcal{W}, K$

Output: $\mathcal{I}, T, B$

1:  for $\mathrm{F}_i \in \mathcal{F}$ do

2:      $\ell_i \leftarrow HB_2; C_i \leftarrow Enc(K_2, \mathrm{F}_i)$

3:      $hash_i \leftarrow HB_1; B[\ell_i] \leftarrow hash_i; T[\ell_i] \leftarrow C_i$

4:  end for

5:  for $\overline{w_j} \in \mathcal{W}$ do

6:      for $id \in DB$ do

7:          $u_{w_j} \leftarrow B_1; st_j \leftarrow B_2; t_{w_j} \leftarrow HB_2$

8:          Extract positions $(pos_1, pos_2, ..., pos_m)$ of keyword $\overline{w_j}$ in file $\mathrm{F}_i$

9:          $SL^i_{w_j} = \pi(id_i) + E(pos_1) + E(pos_2) + ... + E(pos_m)$

10:         $v_{\mathcal{B}} \xleftarrow{R} \{0,1\}^{(l+\lambda)}; \mathcal{I}[t_{w_j}] \leftarrow v_{\mathcal{B}}; \sum[w_j] = st_j$

11:     end for

12: end for

13: send $(\mathcal{I}, B)$ to blockchain, send $(\mathcal{I}, T)$ to cloud server

---

**Algorithm 7**  **Simulator 7.**

Input: $\mathcal{I}, T, TK_{i,Q}$

Output: Search results $R$

1: Parse $\mathbf{ph}(\mathbf{w})$ as $[(t_1, pf_1), (t_2, pf_2), ..., (t_c, pf_c)]$

2: Parse $\{\overline{t_{w_1}}, \overline{t_{w_2}}, ..., \overline{t_{w_t}}, E_1, E_2, ..E_{t-1}\} \leftarrow min\ \mathbf{sp}(\overline{TK_{i,Q}})$

3: for $\overline{t_{w_j}} \in min\ \mathbf{sp}(\overline{TK_{i,Q}})$ do

4:      $v_{\mathcal{B}} \xleftarrow{R} \{0,1\}^{(l+\lambda)}; (\mathcal{B}_{w_j} || SL^i_{w_j}) \xleftarrow{R} \{0,1\}^{(l+\lambda)}$

5: end for

6: $\mathcal{B} = \mathcal{B}_1 \wedge \mathcal{B}_2 \wedge ... \wedge \mathcal{B}_t$

7: Get $ID_{\mathcal{B}} = \{\ell_1, \ell_2, ..., \ell_p\}$ from $\mathcal{B}$

8:  for $\ell_i \in ID_{\mathcal{B}}$ do

9:      $flag = \{000...000\}^{t-1}$

10:     if $\hat{X}[i] = \bot$ then

11:         $\hat{X}[i] \xleftarrow{R} \{0,1\}^n$

12:     else

13:         $\hat{X}[i] \leftarrow pf_i$

14:     end if

15:     $(E(pos^1_1), E(pos^1_2), E(pos^1_m)) \leftarrow SL^1_{w_1}$

16:     for $d = 2 \rightarrow t$ do

17:         $(E(pos^i_1), E(pos^i_2), E(pos^i_n)) \leftarrow SL^i_{w_j}$

---

(Continued)

**Algorithm 7** (**continued**)

18:   for $k = 1 \rightarrow m; k' = 1 \rightarrow n$ do

19:    if $E(pos_1^{k'}) = E(pos_1^k) \times E(d - 1)$ then

20:     Set the position $(i - 1)$ of *flag* to 1

21:     **break**

22:    end if

23:   end for

24:  end for

25: If all positions of *flag* are 1

26: get $c_i \leftarrow T[\ell_i], R \leftarrow R \cup c_i$

27: end for

28: Server sends $\{\hat{X}\}$ to the blockchain for verification, and sends $R$ to the data user

---

**Algorithm 8** **Simulator 8.**

Input: $\mathscr{I}, B, TK_{i,Q}, \psi$

Output: *proof*

1: Parse $\mathbf{ph}(\mathbf{w})$ as $[(t_1, pf_1), (t_2, pf_2), \ldots, (t_c, pf_c)]$

2: $\psi' \leftarrow \phi, proof = 0.$

3: Using search trapdoors $\overline{TK_{i,Q}}$ to perform phrase searches same as line 1–line 6 of Simulator 7.

4: for $\ell_i \in ID_{\mathscr{B}}$ do

5:  if $\hat{X}'[i] = \perp$ then

6:   $\hat{X}'[i] \xleftarrow{R} \{0, 1\}^n$

7:  else

8:   $\hat{X}[i] \leftarrow pf_i$

9:  end if

10:  $hash_i' \leftarrow B[\ell_i], \psi' \leftarrow +_{\mathscr{H}}\mathscr{H}(r, hash_i)$

11: end for

12: if $\psi = \psi'$ then

13:  $proof = 1$

14: end if

15: The blockchain sends *proof* to the data user.

---

with a Linux operating system, Intel Core i7-8550 CPU, and 8 GB RAM. Experimental programs are developed using Python. As for the pseudo-random functions $F$ and the hash function $H$ in the algorithm, we use HMAC-SHA-256 and SHA-256 respectively to implement them. Additionally, we symmetrically encrypt files using AES-128, and the security parameter is set to 128 bits. To evaluate our scheme in practice, we employ the Enron email dataset (*Cukierski, 2015*), a real-world dataset comprising over 517 thousand

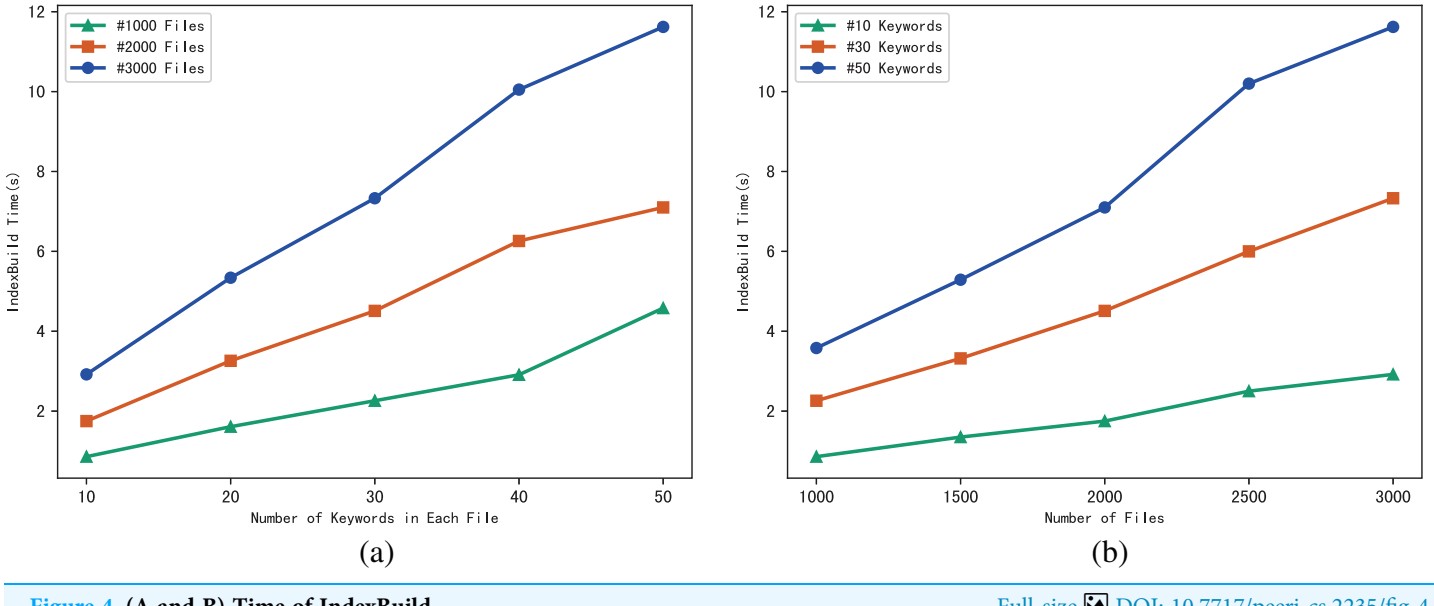

**Figure 4 (A and B) Time of IndexBuild.**

documents. Using the Porter Stemmer, we extract more than 1.67 million keywords and eliminate irrelevant terms such as "of" and "the".

## Evaluation of IndexBuild

In this phase, the IoT device mainly completes the following work: (1) encrypt the files in the system into ciphertext; (2) generate the secure index for all the keywords; (3) calculate checklist of the ciphertext for verification.

The performance of the scheme can be evaluated through the execution time of algorithm IndexBuild, and we evaluate the execution time of IndexBuild in different numbers of files and keywords respectively. Figure 4A shows the variation pattern between the execution time of IndexBuild and the number of keywords in a single file, while files changes from 1,000, 2,000 to 3,000; in contrast, Fig. 4B shows the variation between the execution time of IndexBuild and the number of files in the system, while keywords in a single file changes from 10, 30 to 50. Obviously, the execution time of algorithm IndexBuild is affected by both the number of files and keywords. The more files and keywords contained in each file, the more time it takes in IndexBuild.

## Evaluation of TokenGen

Search trapdoors are generated by users, which contains the permutation value of each keyword in the query phrase and the encrypted distance for other keywords except the first one. Figure 5 shows the time it takes to calculate a search trapdoor for different sizes of search phrases, it's clear that the time increases with the size of the query phrase. This is easy to understand, because the more keywords in the phrase, the more distances between keywords that need to be encrypted, resulting in more trapdoor calculation time.

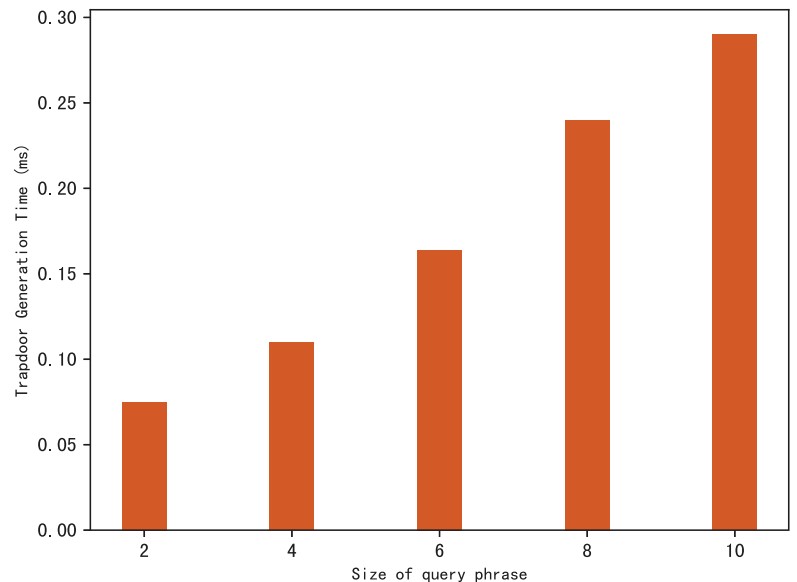

**Figure 5  Time of trapdoor generation.**

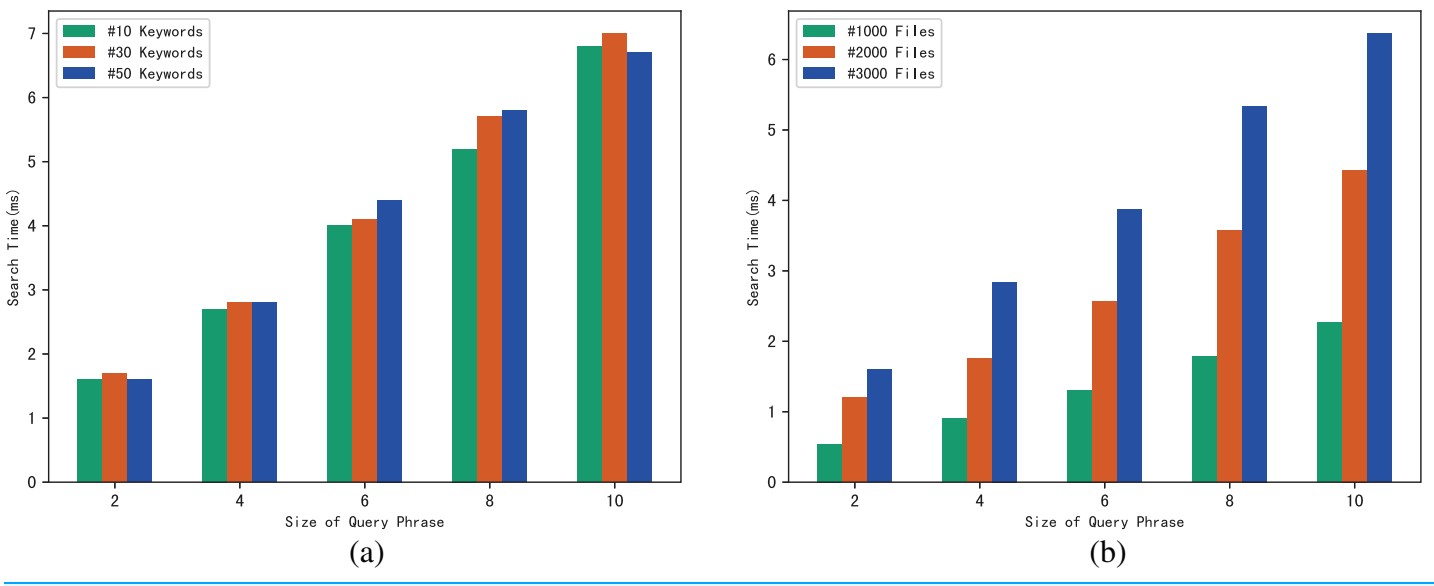

**Figure 6  (A and B) Time of search.**

## Evaluation of search

Similarly, we use execution time to evaluate the search efficiency of our scheme. Figure 6A shows how search time changes with query phrase size when the number of keywords contained in each file is 10, 30 and 50. Figure 6B shows how search time changes with query phrase size when the number of files is 1,000, 3,000 and 5,000.

The results of Fig. 6 demonstrate that as the size of the query phrase grows or the number of documents grows, the search time will increase accordingly, which indicates that the more keywords in the phrase and the more files in the system, the more time it

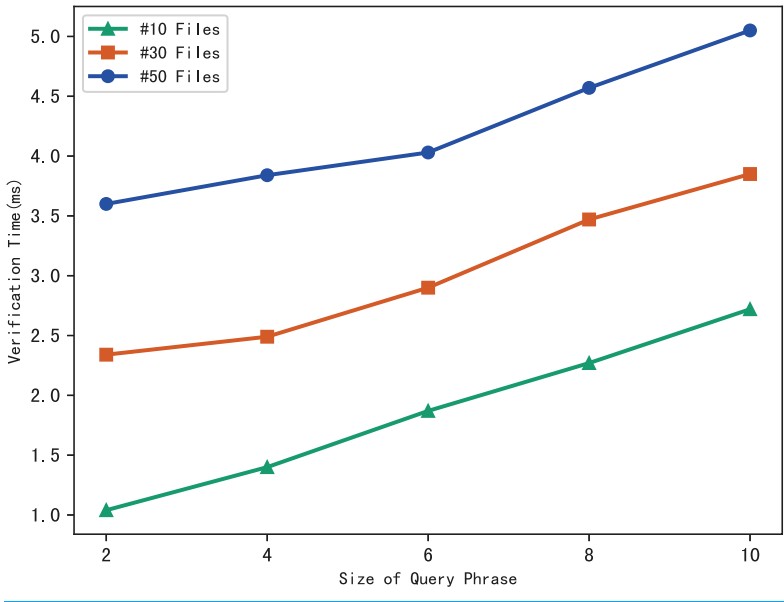

**Figure 7 Time of verification.**

takes for the server to perform a phrase search. Furthermore, since we can use trapdoors to directly locate keywords in the inverted index, the search time is independent of the number of keywords contained in each file.

## Evaluation of verification

The blockchain first performs the similar operations as the server to search files that meet the search phrase, then gets the corresponding hash value through the checklist $B$, and finally calculates the benchmark $\psi'$ based on the multi-set hash function. During the verification, the blockchain draws the verification conclusion by comparing $\psi'$ with the search aggregation evidence $\psi$ returned by the server. Therefore, the verification time is related to the number of files that match the query phrase, and the experimental results are shown in Fig. 7. Clearly, the verification time grows sub-linearly with the number of files and the size of the phrase.

The gas consumption during the verification process is shown in Fig. 8. During the verification process, the blockchain performs multiset hash calculations on the hash values in the checklist that meet that meet the requirements of the phrase search, so the gas consumption increases with the number of search results. When the number of resulting files is 5, the gas consumption is $1.6 \times 10^5$, and when the number of files is 25, the gas consumption is $6.8 \times 10^5$, gas consumption grows sublinearly.

## Comparison with existing schemes

We choose scheme (*Kissel & Wang, 2013*) and scheme (*Ge et al., 2021*) with similar functions to compare, and the results are shown Table 2, in which VPS and VPS-IoT denote scheme (*Kissel & Wang, 2013*) and scheme (*Ge et al., 2021*), respectively. Both VPS and VPS-IoT adopt a two-stage search strategy, so it takes two rounds of interaction between the user and the cloud server to complete a phrase search. Additionally, they

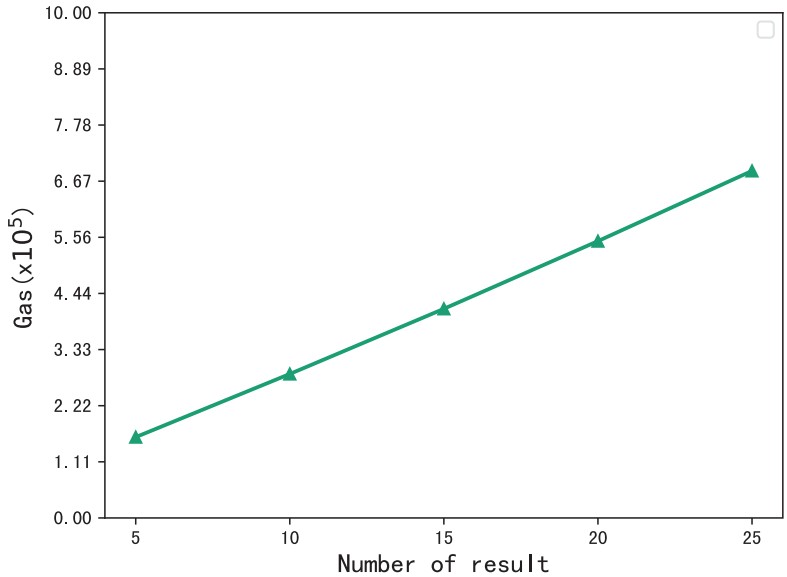

**Figure 8 Gas consumption for verification.**

**Table 2 Compare with existing schemes.**

| Scheme | Index building | Trapdoor generation | Query | Verification | Round |
|---|---|---|---|---|---|
| VPS | $O(MN)$ | $O(|q| * |F_w|)$ | $O(|F_w| * |F_j|^{|q|})$ | $O(|q| * |F_w|)$ | 2 |
| VPS-IoT | $O(MN) + O(\sum_{j=1}^{M} N|F_j|)$ | $O(2|q| + |F_w| + |q| * |F_w|)$ | $O(|F_w| * |F_j|^{|q|} + |q|)$ | $O(|q| * |F_q|)$ | 2 |
| Our Scheme | $O(MN)$ | $O(|q| * |F_w|)$ | $O(|F_w| * |F_j|^{|q|})$ | $O(|F_w|)$ | 1 |

**Note:**
$|F_j|$ is the number of keywords contained in the file $F_j$; $w_j$ is a collection of different keywords in the file $F_j$; $|q|$ is the number of query keywords; $|F_w|$ is the number of files containing query keywords; $|F_q|$ is the size of the returned result file.

calculate the verification evidence by the server. In this case, if the server is not trustworthy, the evidence may also be incorrect, which poses a huge threat to the reliability of verification.

The experimental results are shown in Figs. 9–12. The results in Fig. 9 show that VPS takes the least time in building the index, while VPS-IoT takes the most. The reason lies in that VPS lacks verification of file integrity, so the calculation cost is low. Both our scheme and scheme VPS-IoT can verify the integrity of the file, but the structure of the lookup table in VPS-IoT is complex, requiring a large number of encryption and MAC operations on keyword positions, ciphertext, *etc.*, so it needs more time than our scheme. Figure 10 represents that our scheme gains the highest efficiency in trapdoor generation. Both VPS and VPS-IoT adopt a two-phase query strategy, the data user generates two trapdoors for a query, while our scheme only needs to generate one trapdoor, obviously, our scheme is more efficient. Figure 11 shows the comparison of the query efficiency of the three schemes, the query is performed over 1,000 files and each file contains 20 keywords. The complexity of the three schemes is almost the same, the search time grows sub-linearly with the number of keywords in the phrase. As for verification efficiency, we deploy the

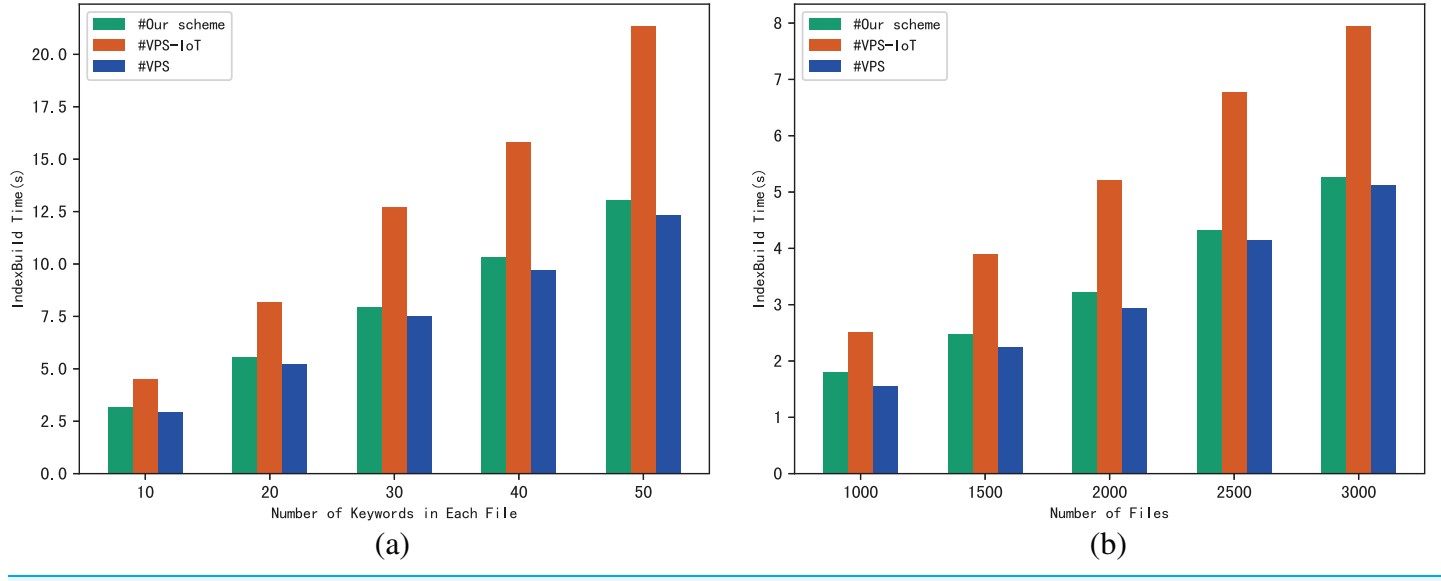

**Figure 9 (A and B) IndexBuild.**

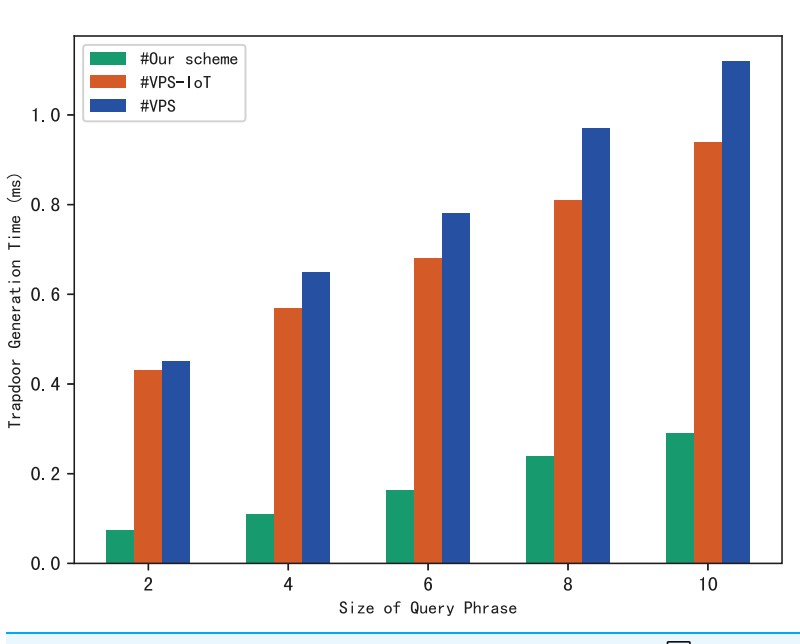

**Figure 10 Trapdoor generation.**

three schemes on 50 files, and the experimental results are shown in Fig. 12. Scheme VPS-IoT performs best among the three schemes, but it cannot verify the integrity of the file. Our scheme takes less time than scheme VPS when the size of the query phrase becomes larger, which demonstrates the efficiency of our scheme. Furthermore, the verification is performed on the blockchain in our scheme, ensuring the reliability of verification.

From what we have discussed above, our scheme has obvious advantages in index construction, trapdoor generation, and result verification compared with existing schemes,

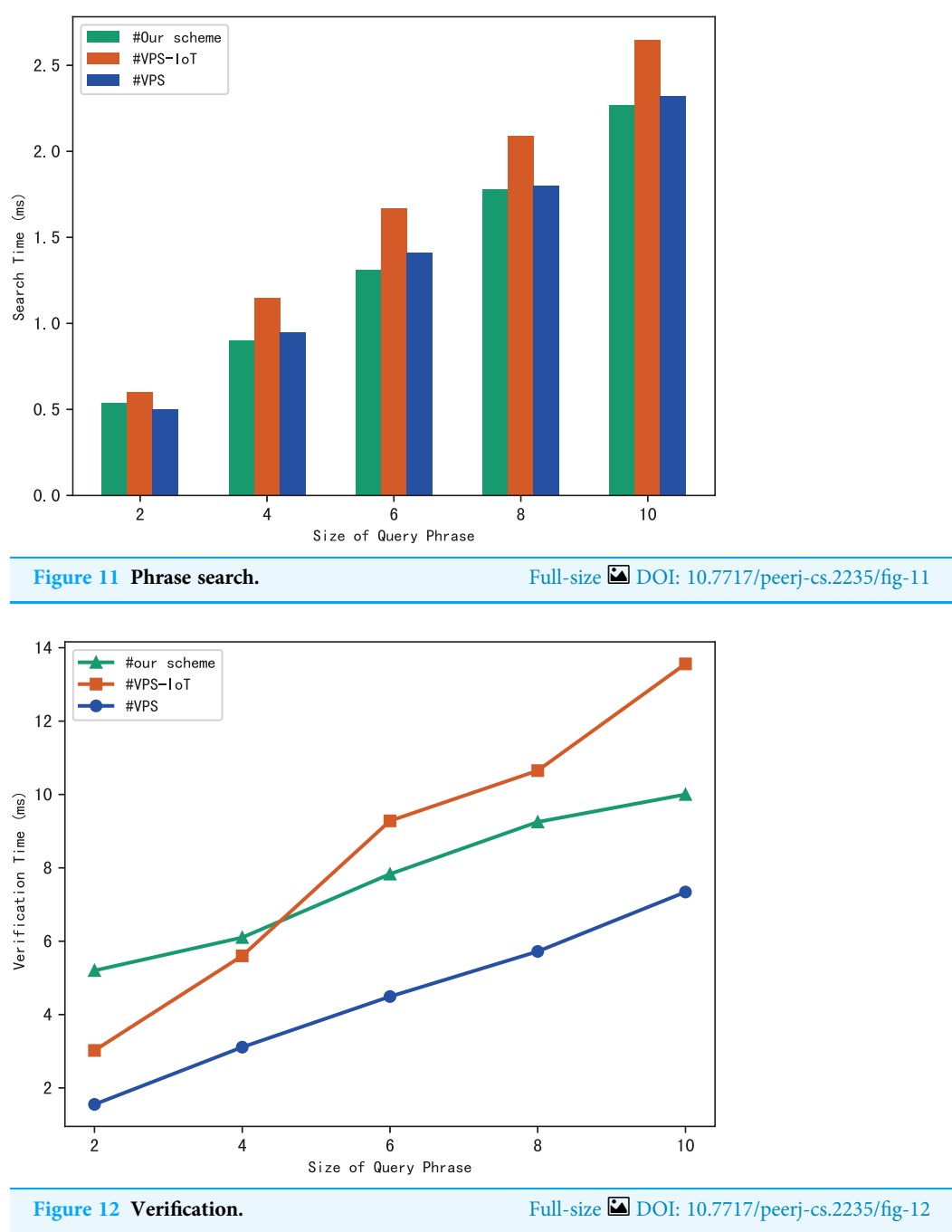

**Figure 11 Phrase search.**

**Figure 12 Verification.**

and the search efficiency is comparable to existing schemes. Furthermore, our scheme enables reliable and complete verification of search results with the help of blockchain, preventing the server from generating unreliable verification evidence due to only storing partial indexes and ciphertexts. At the same time, our scheme can prevent the unfair verification problem caused by malicious users forging verification results.

## DISCUSSION

In this article, we presented a efficient phrase search scheme with reliable verification over encrypted cloud-IoT data, which tackled the challenges of efficient phrase identification and reliable result verification. The scheme introduces the blockchain to the verification which ensures the reliability of the verification evidence and verification process. During the verification process, we use a multiset hash function to aggregate the on-chain evidence into a hash value, which significantly reduces the blockchain transaction cost. In addition, the scheme designs a novel compound Index and distance discrimination algorithm that can quickly determine the order of keywords and achieve efficient identification of phrases, which reduces the computational and communication overhead.

### Funding
This work supported by the National Natural Science Foundation of China (62302467, 62202437, 62201525) and "the Fundamental Research Funds for the Central Universities". The funders had no role in study design, data collection and analysis, decision to publish, or preparation of the manuscript.

### Grant Disclosures
The following grant information was disclosed by the authors:
National Natural Science Foundation of China: 62302467, 62202437, 62201525.
Fundamental Research Funds for the Central Universities.

### Competing Interests
The authors declare that they have no competing interests.

### Author Contributions
- Wanshan Xu conceived and designed the experiments, performed the experiments, analyzed the data, performed the computation work, prepared figures and/or tables, authored or reviewed drafts of the article, and approved the final draft.
- Ze Zhu conceived and designed the experiments, performed the experiments, performed the computation work, prepared figures and/or tables, and approved the final draft.
- Muhammad Irfan Khalid analyzed the data, authored or reviewed drafts of the article, and approved the final draft.

### Data Availability
The code is available in the Supplemental File.
The Enron Email Dataset is available at: http://www.cs.cmu.edu/~enron/.

### Supplemental Information
Supplemental information for this article can be found online at http://dx.doi.org/10.7717/peerj-cs.2235#supplemental-information.

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
