# Peer review of "Efficient phrase search with reliable verification over encrypted cloud-IoT data"

_PeerJ Computer Science, doi:10.7717/peerj-cs.2235_

## Round 0.1 · original submission · Major Revisions

The manuscript explores a meaningful topic in the realm of cloud storage, namely searchable encryption, yet several crucial aspects require improvement. Firstly, the research work lacks thoroughness in highlighting differences from previous studies and overlooks key literature, such as a reference employing a multiset hash function for verifiability of search results, diminishing its academic value. Secondly, the organizational structure and writing standards need refinement to enhance readability and clarity. Thirdly, ambiguities in algorithm syntax and security definitions hinder comprehension, while the security analysis section lacks formal rigor. Moreover, questions arise regarding the motivation and contributions of the work, the practicality of the system model, and the role of blockchain immutability in ensuring reliable search results. Addressing these concerns will bolster the manuscript's academic rigor and contribution to the field.

Reviewer 1 ·

Basic reporting

In the context of cloud storage, searchable encryption has significant practicality, and the topic discussed in this manuscript is very meaningful. However, there are still several aspects that need to be improved.
1. The research work of the manuscript is slightly insufficient, failing to highlight the differences from previous studies, and missing some key literature, such as the reference with DOI: 10.1109/TCC2022.3170362.

In this scheme, the authors also employed a multiset hash function to achieve verifiability of search results. The insufficient research in the manuscript has to some extent weakened its academic value and persuasiveness.
2. The organizational structure of the manuscript still needs to be optimized, lacking clear section arrangements and content introductions, making it difficult for readers to quickly grasp the context and key points of the article while reading.
3. The writing standards of the manuscript also need to be improved. For example, using some abbreviations is not standardized enough, which may confuse readers in understanding.
4. In terms of algorithm syntax definition, the sources of K_2 and K_3 are not clearly explained, which increases the difficulty for readers to understand the algorithm.
5. In the security definition section, the definition of the leakage function is missing, resulting in a lack of the necessary theoretical basis for security analysis.
6. The security analysis section of the manuscript appears to be not formal enough, lacking rigorous mathematical derivation and proof, which affects the persuasiveness of the article.
7. The format of references is not uniform, and some references lack key information such as page numbers, which to some extent affects the academic rigor of the manuscript.

Experimental design

no comment

Validity of the findings

no comment

Cite this review as

Reviewer 2 ·

Basic reporting

no comment

Experimental design

no comment

Validity of the findings

no comment

Additional comments

1. Unclear motivation and contributions. This work is motivated by the need to verify the search results, and one of the main contributions is to enable reliable verification of phrase search results, as stated by the authors. However, the recent proposal [9] has enabled the data user to verify the phrase search results even when the server is malicious and tries to fabricate the verification. What is the difference between [9] and this work?

2. Impractical system model. In the defined model, the blockchain is responsible for performing computation for a phrase search in parallel. It generates the verification value in order to validate the search results generated by the server. This requires a certain computational capability, which the blockchain is not assumed to have in a common setting. The blockchain in the defined model is actually a trusted entity needed to perform computational tasks, which is impractical in the real world. Besides, the authors consider the data user to be not trustworthy as it may falsify verification results for financial gain. Why the data user can benefit from it?

3. Another question. The authors say that due to the immutability of data on the blockchain, the composite index stored and search results calculated by the blockchain are reliable. Why the immutability of blockchain can ensure that the search results are reliable? Isn't the reliability of the results guaranteed by the algorithm?

Cite this review as

---

## Round 0.2 · Minor Revisions

The manuscript shows potential but needs revision to address the identified issues. A comprehensive review focusing on grammatical correctness, consistency in verb tenses, and the refinement of expressions is necessary to elevate the manuscript's quality.

Reviewer 1 ·

Basic reporting

Although the authors have made various optimizations and improvements in this version, the manuscript still faces some unresolved issues:
1. There are still some grammatical errors in the manuscript, such as line 37. To ensure the delicacy and rigor of the manuscript, the authors need to re-review the entire text to improve its overall quality.
2. The use of tenses in the manuscript appears to be quite chaotic, such as the present tense on line 117 and past tense on line 119 within the related work section. It is recommended that the authors carefully review the entire manuscript again to ensure consistency and accuracy in verb tense usage.
3. There is still room for improvement in certain expressions in the manuscript, such as line 33. To ensure the manuscript's high quality, the authors should re-check the manuscript.

Experimental design

no comment

Validity of the findings

no comment

Cite this review as

Reviewer 2 ·

Basic reporting

no comment

Experimental design

no comment

Validity of the findings

no comment

Additional comments

The authors have addressed the concerns I raised.

Cite this review as

---

## Round 0.3 · accepted · Accept

The reviewer agreed to accept the paper and I agree with them

Reviewer 1 ·

Basic reporting

no comment

Experimental design

no comment

Validity of the findings

no comment

Cite this review as